# Gene Expression, Histology and Histochemistry in the Interaction between *Musa* sp. and *Pseudocercospora fijiensis*

**DOI:** 10.3390/plants11151953

**Published:** 2022-07-27

**Authors:** Julianna Matos da Silva Soares, Anelita de Jesus Rocha, Fernanda dos Santos Nascimento, Vanusia Batista Oliveira de Amorim, Andresa Priscila de Souza Ramos, Cláudia Fortes Ferreira, Fernando Haddad, Edson Perito Amorim

**Affiliations:** 1Departamento de Ciências Biológicas, Universidade Estadual de Feira de Santana, Feira de Santana 44036-900, BA, Brazil; juliannamatos91@gmail.com (J.M.d.S.S.); anelitarocha@gmail.com (A.d.J.R.); feel.20@hotmail.com (F.d.S.N.); 2Embrapa Mandioca e Fruticultura, Cruz das Almas 44380-000, BA, Brazil; vanusiaamorim50@gmail.com (V.B.O.d.A.); andresa.ramos@embrapa.br (A.P.d.S.R.); claudia.ferreira@embrapa.br (C.F.F.); fernando.haddad@embrapa.br (F.H.)

**Keywords:** banana, black Sigatoka, gene expression, RT–qPCR

## Abstract

Bananas are the main fruits responsible for feeding more than 500 million people in tropical and subtropical countries. Black Sigatoka, caused by the fungus *Pseudocercospora fijiensis*, is one of the most destructive disease for the crop. This fungus is mainly controlled with the use of fungicides; however, in addition to being harmful to human health, they are associated with a high cost. The development of resistant cultivars through crosses of susceptible commercial cultivars is one of the main focuses of banana breeding programs worldwide. Thus, the objective of the present study was to investigate the interaction between *Musa* sp. and *P. fijiensis* through the relative expression of candidate genes involved in the defence response to black Sigatoka in four contrasting genotypes (resistant: Calcutta 4 and Krasan Saichon; susceptible: Grand Naine and Akondro Mainty) using quantitative real-time PCR (RT–qPCR) in addition to histological and histochemical analyses to verify the defence mechanisms activated during the interaction. Differentially expressed genes (DEGs) related to the jasmonic acid and ethylene signalling pathway, GDSL-like lipases and pathogenesis-related proteins (PR-4), were identified. The number and distance between stomata were directly related to the resistance/susceptibility of each genotype. Histochemical tests showed the production of phenolic compounds and callosis as defence mechanisms activated by the resistant genotypes during the interaction process. Scanning electron microscopy (SEM) showed pathogenic structures on the leaf surface in addition to calcium oxalate crystals. The resistant genotype Krasan Saichon stood out in the analyses and has potential for use in breeding programs for resistance to black Sigatoka in banana and plantains.

## 1. Introduction

Banana is the fourth most important staple food in the world, behind rice, wheat and corn [1,2]. Global production of bananas and plantains in 2020 was approximately 119.8 million tons, with plantations grown on 5.2 million hectares [3]. Four countries account for 48% of global banana production, namely, India (31.5 million tons), China (11.5 million tons), Indonesia (8.1 million tons) and Brazil (6.6 million tons) [3].

Despite the large scale of banana production, this crop is constantly affected by abiotic and biotic factors that directly affect fruit production, including water deficit and temperature extremes, as well as pests and diseases [4]. The main biotic factors affecting banana include bacterial wilt [5], nematodes [6], Fusarium wilt [7], yellow Sigatoka [8] and black Sigatoka [9,10]. Black Sigatoka is thought to have the greatest economic impact since the incidence of this fungus in banana-producing regions has increased significantly in recent years, partly due to climate changes that have made the temperature and humidity conditions favorable for the germination and growth of fungal spores [11].

Black Sigatoka disease (BSD) is caused by *Mycosphaerella fijiensis* M. Morelet (asexual phase: *Pseudocercospora fijiensis* (M. Morelet) Deighton) and affects bananas and plantains worldwide. The disease was first reported in the Sigatoka Valley in the Fiji Islands during the 1960s and has since spread to almost all banana production areas [12]. The fungus *P. fijiensis* directly limits the photosynthetic area of the leaves of the plant. The symptoms are manifested as striae and, in more advanced stages, lead to total necrosis of the leaves, causing early ripening of the fruits and significant economic loss, which can reach up to 100% of the yield, depending on the cultivar, environmental conditions, and the level of infection [1,13,14].

The disease is mainly controlled by limiting contact and the use of systemic fungicides, which are applied with a relatively high frequency; this has significantly increased production costs, which are estimated at more than 550 million dollars annually [1,15]. In many areas, the disease leads to a low availability of fruit for local consumption, increasing prices for consumers. In addition, small producers have difficulty controlling the disease due to the high cost of fungicides, and potential risk to human health and environmental damage should also be considered [1,15].

The banana export market, based on cultivars of the Cavendish subgroup, faces several production limitations. Black Sigatoka has a prominent role in these limitations given the high susceptibility of the commercial cultivars. Therefore, to maintain productivity and meet the demands of consumers, especially in Europe, strict control measures are deemed necessary. However, restrictions on the use of chemicals in agriculture have also been imposed in Europe, prohibiting some active ingredients; the fungicides routinely used to control black Sigatoka, such as mancozeb, have been affected by these restrictions [16,17]. As a result, there is an urgent need for alternatives for disease and genetic control, therefore, the use of resistant cultivars becomes the most promising solution [18,19,20].

Banana Genetic Breeding Programs, especially the Fundación Hondureña de Investigación Agrícola (FHIA, Honduras), the Center de coopération internationale en recherche agronomique pour le développement (CIRAD, France), the International Institute of Tropical Agriculture (IITA, Nigeria), the National Research Center for Banana (NRCB, India), and the Brazilian–Empresa Brasileira de Pesquisa Agropecurária (Embrapa, Brazil), focus on developing cultivars resistant to black Sigatoka. These programs promote crosses between commercial and diploid cultivars and the selection of promising genotypes within the progenies as the main breeding method [21].

Molecular analyses of the interaction between *Musa* sp. and *P. fijiensis* have the potential to provide useful information about the complex of genes involved in pathogen responses, allowing the use of this knowledge in the design of molecular markers for assisted selection or even for genetic engineering studies, accelerating the development of cultivars resistant to black Sigatoka. However, information in the literature on genes involved in the interaction between *Musa* sp. × *P. fijiensis* remains scarce; thus, more information pertaining this pathosystem is needed [21].

The objective of this study was to determine and validate the expression profiles of the candidate genes for resistance found in the literature, using RT–qPCR. Two subtractive suppression hybridization (SSH) cDNA libraries were identified for Calcutta 4 and Grand Naine after infection by *P. fijiensis*; these cultivars have contrasting genotypes regarding resistance and susceptibility to the pathogen [22,23]. Additional histological and histochemical analyses were performed in order to identify which other defence mechanisms are involved in the *Musa* sp. × *P. fijiensis* interaction.

## 2. Results

The progression of symptoms and confirmation of the presence of the pathogen *P. fijiensis*, (isolate collected from the cultivar “Grand Naine”) in the inoculated plants was verified at 45 days after inoculation (DAI), as shown in Figure 1. The symptoms were observed on the abaxial surface of the leaves. The resistant genotypes Calcutta 4 and Krasan Saichon showed no symptoms. However, in the susceptible genotypes Grand Naine and Akondro Mainty, there was an increase in leaf spots over the inoculation period, with greater severity beginning at 15 DAI. The disease progressed as expected, with brown streaks that increased in size following inoculation.

### 2.1. Analysis of Gene Expression

The ACC oxidase gene [23] showed increased expression in the resistant genotypes Calcutta 4 and Krasan Saichon compared to that in the Grand Naine genotype and remained so throughout the evaluation period. However, among the susceptible genotypes, there was an overexpression only in the Akondro Mainty genotype starting at 15 DAI, while for Grand Naine, the gene was downregulated at 9 DAI (Figure 2A).

For all the genotypes there was positive expression of the GDSL-like lipase gene [23]; however, the highest level of expression occurred in Akondro Mainty starting at 9 DAI, with an increase in expression until 21 DAI (Figure 2B).

The JAR1 gene [23] showed progressively greater expression up to 21 DAI in the resistant genotypes, and downregulated in the susceptible genotypes at 3 DAI for the Grand Naine genotype and at 15 DAI for the Akondro Mainty genotype with upregulation at 21 DAI (Figure 2C).

The PR-4 gene [23] was expressed progressively in all the genotypes starting at 3 DAI. This gene was upregulated in Calcutta 4 and Krasan Saichon in comparison to Grand Naine, mainly at 3 and 9 DAI. In Akondro Mainty, there was a higher expression of the PR-4 gene than in the other genotypes at all times. In Grand Naine, this gene remained stable throughout the entire period evaluated (Figure 2D).

For the LOX gene [24], there was positive expression in all the genotypes except for Grand Naine at 3 DAI, in which this gene was negatively regulated. A gradual expression of the resistance genes was observed for all the genotypes. There was overexpression in the contrasting genotypes Krasan Saichon and Akondro Mainty starting at 9 DAI (Figure 2E).

The overall expression of the five selected genes is shown in the heat map in Figure 3. The analyses show the hierarchical grouping of the genotypes Calcutta 4, Krasan Saichon, Grand Naine and Akondro Mainty according to the expression profiles of the five genes analyzed. The dendrogram lines represent the genotypes and the collection times, and the columns represent the selected resistance genes.

In general, for the global gene expression profiles, the Akondro Mainty genotype had a higher expression level for all the genes in study. However, late expression was observed for this genotype. For the resistant cultivars (Calcutta 4 and Krasan Saichon), 3 DAI was considered the optimal time, reflecting rapid expression after inoculation with *P. fijiensis*.

### 2.2. Histological Analyses

#### 2.2.1. Leaf Clarification and Staining

Figure 4 shows the images taken after leaf clarification and trypan blue staining for all the genotypes studied. In the samples taken at 21 DAI, there was colonization of the pathogen in the leaf tissues near the stomatal structures that *P. fijiensis* uses to penetrate the plant. For the control treatments (plants not inoculated), no pathogen structures were observed (Figure 4A–D).

Hyphae and pathogen structures were found in all the samples from inoculated plants. In Calcutta 4, there were hyphae and ascospores present; however, these structures were at distant and superficial points of the leaf and in smaller amounts (Figure 4E). In Krasan Saichon, only hyphae were present without any other reproductive structures, with penetration of the pathogenic hyphae into the stoma (Figure 4F). In Grand Naine, a tangle of hyphae that extended internally throughout the tissue was observed at many points on the leaf surface (Figure 4G). In Akondro Mainty, reproductive structures of the pathogen, such as ascospores and hyphae scattered throughout the tissue, were also noticed (Figure 4H).

#### 2.2.2. Stomatal Density

Results of the Tukey test at 5% was significant for the differences between the resistant and susceptible genotypes studied regarding the number of stomata and the distance between them (Figure 5C). The density of the stomata was lower in the resistant genotypes, with an average of approximately 38 stomata/mm^2^ for Calcutta 4 (Figure 5D,E) and 17 stomata/mm^2^ for Krasan Saichon (Figure 5F,G). Among the susceptible genotypes, the average values were 70 stomata/mm^2^ and 71 stomata/mm^2^, for Grand Naine (Figure 5H,I) and Akondro Mainty (Figure 5J,K), respectively. Regarding the distance between the stomata (Figure 5B), there was a significant difference between the resistant genotypes, with Krasan Saichon having the largest distance between stomata; there was no significant difference in stomata distance between the susceptible genotypes. However, a greater distance between stomata was seen in Akondro Mainty in comparison to Grand Naine (Figure 5B).

#### 2.2.3. Histochemical Analysis

Figure 6 shows the images used to detect phenolic compounds which are present in the cells of the palisade parenchyma and spongy parenchyma following the infectious process. The ferric chloride used produces a black or reddish-brown color in the presence of phenolic compounds. The amount of phenolic compounds increased gradually over the interaction time, being more evident at 21 DAI in the Calcutta 4 and Krasan Saichon genotypes (Figure 6C,F). Phenolic compounds were present in the genotypes Akondro Mainty and Grand Naine at 21 DAI (Figure 6I,L).

Figure 7 shows the images used for the detection of callose, which were identified by fluorescence after staining with aniline blue + Lugol. The production of this compound was observed in the resistant genotypes at 21 DAI (Figure 7C,F).

Figure 8 shows the results of the SEM (scanning electron microscopy) analysis performed for the resistant (Calcutta 4 and Krasan Saichon) and susceptible (Grand Naine and Akondro Mainty) genotypes. The absence of the pathogen in the non-inoculated samples (Figure 8A,E,I,M), confirmed that there was no contamination in the control treatment. The presence of hyphae on the leaf surface was evident at 9 DAI (Figure 8B), and at 21 DAI, calcium oxalate crystals, which have characteristics similar to a needle and are known as raphides, were detected in the Calcutta 4 cultivar (Figure 8C). In the Krasan Saichon cultivar, hyphae and calcium oxalate crystals were observed at 21 DAI. Although the crystals were in the shape of prisms, they still presented the geometric characteristics (Figure 8G). In the susceptible cultivars, wax was seen on the leaf surface. No hyphae were detected in either resistant cultivar in the inner part of the leaves in the cross-sections (Figure 8D,H).

In the Grand Naine genotype (Figure 8I–L), hyphae were observed at 9 DAI and 21 DAI (Figure 8J,K). At 21 DAI, calcium oxalate crystals were observed in a typical prismatic pattern around the stomata, and hyphal structures were observed penetrating the stomata (Figure 8K). In the Akondro Mainty genotype (Figure 8M–P), hyphae were observed at 9 and 21 DAI (Figure 8N,O). In both susceptible genotypes, hyphae were detected in the inner part of the leaves from the cross-sectional views (Figure 8L,P).

## 3. Discussion

### 3.1. Gene Expression

In this study, five genes were evaluated as possible candidate genes for resistance to *P. fijiensis* in Calcutta 4, Krasan Saichon, Grand Naine and Akondro Mainty. The expression of genes related to plant defence showed different results among the contrasting genotypes studied. This is the first study that analyzes the gene expression pattern in the interaction of the fungus *P. fijiensis* with the genotypes Krasan Saichon (resistant) and Akondro Mainty (susceptible).

The importance of this study for the Krasan Saichon genotype is that this genotype is resistant to black Sigatoka, one of the main fungal diseases of bananas [9]. Thus, this accession can be used in crosses with other diploids to obtain improved diploids with superior characteristics. It is also important to have available improved diploids with different genetic backgrounds in order to broaden the genetic basis of the commercial germplasm and, therefore, hasten the development of banana varieties resistant to diseases, such as black Sigatoka [25].

When plants are infected by pathogens, a large number of genes are involved in a cascade of transcriptional regulation, signal transduction, metabolic activities and defence responses [26].

The JAR1 and LOX genes used in this study are related to jasmonic acid (JA) signalling and therefore have been used as marker genes for this pathway in several molecular studies [27]. In our study, the JAR1 and LOX genes were upregulated in the resistant genotypes Calcutta 4 and Krasan Saichon starting at 3 DAI, while there was a downregulation in the susceptible genotypes Grand Naine (3 DAI) and Akondro Mainty (15 DAI). Our results for the Grand Naine genotype do not corroborate those of Portal et al. [23], who observed stable expression of the JAR1 gene up to 23 DAI. Another study found increased expression of this gene in the susceptible cultivar Williams 72 h after inoculation [24]. One of the main roles of jasmonic acid is the defence mechanism of plants in response to injuries and attacks by pests and pathogens [28,29], and after challenge by stress, there is a rapid increase in the levels of JA [30]. In this study, the JAR1 gene, responsible for the activation of the jasmonic acid signalling pathway, was expressed from the first days after inoculation in the susceptible genotypes.

JA is classified as a cyclopentane fatty acid and is biosynthesized from linolenic acid, one of the main fatty acids of plant cell membranes; all the details of this pathway have been well studied [31,32]. JA and its derivatives, including methyl ester (MeJA) and its isoleucine conjugate (JA-Ile), are called jasmonates (JAs) [32]. The regulation of genes related to phytohormones that mediate defence responses is not fully understood in *Musa* sp., although the signalling associated with jasmonic acid (JA), salicylic acid (AS) and ethylene (ET) also participates in defence responses against pathogens [21,23]. It is believed that the signalling associated with JA and ET triggers resistance against necrotrophic pathogens, while SA activates resistance responses against biotrophic and hemibiotrophic pathogens [33]. Although the JA signalling pathway has been widely investigated, its behavior under different environmental stresses is still uncertain due to the complexity between the various signalling pathways [32,34]. However, in our study, it is clear that the JAR1 and LOX genes were responsible for mediating responses against environmental stresses through JA activation.

The ACC oxidase gene is responsible for catalyzing the last step of ethylene synthesis [23]. In the present study, there was a repression of this gene in the susceptible genotype, Grand Naine at 9 DAI; however, beginning at 15 DAI, there was positive regulation, corroborating the results found by Portal et al. [23], who identified stability in the expression of this gene, up to 23 DAI. The resistant genotype Akondro Mainty showed an expression twice as large as that of Grand Naine.

Ethylene (ET) is a simple gaseous hydrocarbon. It is an important plant hormone that has profound effects on plant development and responses to environmental stimuli [35,36]. Ethylene biosynthesis is controlled by complex mechanisms of transcriptional and posttranscriptional biosynthesis and is formed from methionine through the sequential enzymatic activity of S-adenosyl-methionine (SAM), 1-carboxylic acid-1-aminocyclopropane (ACC) and ACC oxidase (ACCO) [35,36]. High levels of ET are produced in response not only to herbivory but also to mechanical damage [37]. However, the molecular mechanisms responsible for the initial recognition of the pathogen that subsequently activates ET-mediated responses are still poorly understood.

Plants have physical, chemical and molecular barriers to protect themselves from pathogen attack. At the molecular level, in addition to the cellular components that activate the recognition system, molecular patterns associated with microorganisms/pathogens (MAMPs and PAMPs) through pattern recognition receptor proteins (PRRs), pathogenesis-related proteins or (PRs) also play a crucial role in the defence of plants against pathogens. These proteins, comprising a heterogeneous class of plant disease defence proteins, are linked to nucleotides and domains rich in leucine (NLR) and are encoded by R genes [38,39,40]. PRs can sense the presence of effector molecules through specific receptors, initiating the signalling cascade that is characterized by the accumulation of reactive oxygen species (ROS), the biosynthesis of signalling hormones such as AS, JA, and ET, and the induction of defence gene expression [38,41].

The PR-4 gene encodes a protein (chitinase) related to pathogenesis and was induced immediately after inoculation of the pathogen for all the genotypes studied. The results obtained in this study agree with those reported by Rodriguez et al. [42], in which, after 72 HAI, positive regulation was observed in the resistant genotype Calcutta 4, and with the studies conducted by Portal et al. [23], who observed that in the Grand Naine, the susceptible genotype, the expression of PR-4 happened at later time periods. The PR-4 proteins are the first line of the plant defence response and are able to break the fungal cell walls and produce chitin oligomers, triggering plant defence signalling pathways [43,44].

Transcriptomic studies have shown that PR genes are considerably induced by biotic and abiotic stresses, making them one of the most promising candidates for the development of multiple varieties of stress-tolerant crops [45,46,47]. In previous studies, transgenic plants that overexpressed the PR4 genes showed greater resistance to water stress and to infection by *Magnaporthe grisea* in rice [45] and to powdery mildew in grapevines [46].

GDSL-like lipases are related to proteins with antifungal action and are involved in the mechanisms of resistance and susceptibility to pathogens. In this study, the expression of GDSL-like lipases was observed from 3 DAI until 21 DAI in Grand Naine and Akondro Mainty, where there was an overexpression at 15 DAI. However, our results did not corroborate those reported by Portal et al. [23], who observed overexpression of this gene only in Grand Naine at 30 DAI. In contrast, our results were similar to those observed by Concepción-Hernández et al. [48] in a study conducted on the Grand Naine genotype, with expression of this gene at 3 DAI. The balance of ROS and antioxidant enzymes needs to be efficient in plant x pathogen interactions to have a satisfactory resistance response. In our case, similar to the result obtained by Concepción-Hernández et al. [48], this balance was probably not reached, resulting in susceptibility in the Grand Naine genotype.

GDSL-like lipases form a large family of genes, and the expression characteristics of these genes is complex. Some genes are widely expressed at various stages of plant development, while other genes can be expressed only in specific tissues [49]. In addition, the expression of these genes can be induced by stimuli such as biotic and abiotic stress and by phytohormones [50,51], suggesting variability in functions within this group of enzymes. Therefore, the functions of GDSL-like lipases in plant–pathogen interactions can vary greatly between species [49].

Despite the great advances to date, additional functional gene analyses are needed to validate candidate genes for resistance in susceptible banana cultivars [52]. The Krasan Saichon and Akondro Mainty genotypes belong to the Germplasm Bank of Embrapa Mandioca e Fruticultura and in a previous field phenotyping study, they were characterized as resistant and susceptible to black Sigatoka, respectively [9]. In the literature, there are no reports regarding gene expression in response to black Sigatoka in these genotypes; however, the positively regulated gene expression contributes to the understanding of these defence pathways in *Musa* sp., as reported by Portal et al. [23]. Further studies are needed to define the molecular mechanism of the response to black Sigatoka exhibited by Krasan Saichon and Akondro Mainty.

### 3.2. Histological Analyses

This is the first study using clarification and leaf staining technique with trypan blue to analyze the pathogenic structures of *P. fijiensis* in *Musa* sp. (Figure 4). This technique allowed the visualization of fungal structures in their sexual form, with propagation occurring through ascospores (Figure 4E,H). The results of this study show that banana genotypes with higher mean stomatal densities were more susceptible to black Sigatoka. The highest stomatal densities were found in the Grand Naine and Akondro Mainty genotypes, while the lowest were found in the resistant Calcutta 4 and Krasan Saichon genotypes (Figure 5). High stomatal density has been considered a characteristic of cultivars susceptible to black Sigatoka because it can facilitate the penetration of the fungus into the leaf tissue [53,54].

A study on stomatal density in banana genotypes susceptible and resistant to black Sigatoka obtained results similar to those reported in the present study [55,56]. Studies conducted with other pathogens in other crops show a high correlation between stomatal density and increased disease incidence [55,57,58,59]. Thus, the results of this study reaffirm that stomatal density may be related to pathogen penetration because the resistant cultivars Calcutta 4 and Krasan Saichon showed low stomatal density and a lower incidence of symptoms (Figure 1). Certainly, this may be one of the strategies that plants have developed to avoid penetration and consequently further development of a pathogen. However, previous studies that evaluated the morphological and physiological characteristics of leaf stomata indicated that resistance to black Sigatoka in *Musa* sp. may be based on nonstomatal mechanisms, such as host resistance to phytotoxins produced by the pathogen or fungitoxic activity of polyphenols in healthy tissues of resistant genotypes [60].

The results for the analysis of phenolic compounds showed the presence of this substance in greater quantities in the resistant genotypes (Figure 6A,F). In the susceptible genotypes, this compound was present in small quantities at 21 DAI. According to Vermerris and Nicholson [61], phenolic compounds are divided into preforms, which are synthesized during normal plant growth but not at a concentration sufficient to kill the pathogen, and induced compounds, which are secreted during the contact of the pathogen with the host.

Thus, the production of the phenolic compounds that were visualized in this study in the susceptible cultivars Grand Naine and Akondro Mainty may not have been a sufficient defence mechanism to indicate resistance in the interaction between *Musa* sp. and *P. fijiensis*. In the resistant genotypes, Calcutta 4 and Krasan Saichon, the amount of this compound throughout the interaction until 21 DAI was as expected (Figure 6C,F). These compounds are produced and accumulate at a faster rate after infection in resistant varieties than in susceptible varieties [62]. The data obtained corroborate the results found by Torres et al. [63] and Hidalgo et al., 2016 [64], who observed the accumulation of phenolic compounds in resistant banana genotypes.

The presence of callose was observed only in the resistant genotypes, Calcutta 4 and Krasan Saichon (Figure 7C,F). Callose is a common physiological response to different abiotic and biotic stresses [65], including exposure to metals [66], injuries [67] and pathogen attack [68]. In our study, the histochemical analyses suggested that the plants seemed to respond to fungal attack with the production of phenolic compounds and callose.

In the SEM analysis, waxes, hyphae and calcium oxalate (CaOx) crystals were observed on the surfaces of leaves in all the genotypes (Figure 8). The cuticle is formed by cuticular waxes and covers the aerial organs of plants, acting as a protective barrier against environmental challenges such as dehydration, UV radiation, mechanical damage and even the entrance of pathogens [69,70]. Cuticular waxes have gained increasing attention in the study of plant resistance to diseases [71]. In addition to wax, calcium oxalate crystals are also abundant in nature and can be found in more than 215 plant families [72]. Several functions have been attributed to plant CaOx and depend on the quantity, distribution and morphology of the crystals as well as on the characteristics of the cells that produce them, whose main functions are the regulation of high-capacity calcium (Ca) and protection against herbivory [73]. These crystals are stored inside vacuoles of specialized cells called crystal idioblasts [74]; while being broken down in plants, they can produce reactive oxygen species with hydrogen peroxide (H_2_O_2_), which was observed in response to infection by pathogens and is related to the inhibition of infection [75]. In our study, in the resistant genotypes, two different forms of CaOx crystals were seen near the stomata. A study conducted with lily showed that the presence of calcium oxalate crystals in the leaves is a constitutive defence mechanism [76]. On the other hand, other authors reported that in other plant species, the formation of oxalate calcium seems to be an inducible defence response after infection [77,78,79].

The formation of appressoria or other infection structures was not observed in this study, indicating that the main means of entry of the pathogen is through the stomata. Appressoria are specialized infection cells that many pathogenic fungi develop. These cells are able to break the plant surface through physical force and release enzymes that digest the cuticle and the cell wall of the plant [80,81].

In the resistant cultivars, hyphae did not grow on stomatal cells in any of the time periods (Figure 8), and only guard cells were observed laterally; however, in the susceptible genotypes (Figure 8C), hyphae grew on and inside the stomatal cells. Therefore, the findings of our study corroborate previous results suggesting that the penetration of the fungus *P. fijiensis* occurs via stomata [42], usually at 3 DAI. Previous studies of *P. musae* have reported that penetration was observed through stomata between 4 and 6 DAI [82,83]. Thus, it is evident that the observation of hyphae on plant material is not sufficient evidence of infection; a sufficient period is needed for penetration via stomata to occur, completing the infection process. 

Research on plant-fungal interactions from the biochemical and histological perspective has helped improve the understanding of the mechanisms that drive the infectious process [84]. In addition, the development of an effective control of the disease does not depend only on extensive knowledge of the fungal life cycle, and it is of paramount importance to know and understand which resistance mechanisms are used by the host [10]. Thus, histological and histochemical studies allowed the characterization of the process of infection with black Sigatoka in the cultivars studied, and it was possible to observe the production of waxes, calcium oxalate and compounds produced in response to infection by the pathogen. In addition, stomatal density under certain bioclimatic and cultivation conditions may be one of the important factors in regulating the entrance of the fungus into the plant.

The genes included in this study showed similar upregulated expression profiles in the Krasan Saichon genotype inoculated with *P. fijiensis* and in the Calcutta 4 genotype, which makes this genotype promising for use in crosses in banana and plantain breeding programs. In contrast, the susceptible genotype Akondro Mainty showed a different behaviour in gene expression compared to Grand Naine, and further studies are needed to understand the expression levels observed during the interaction. However, in previous studies, the Akondro Mainty genotype was identified as one of the possible ancestors of the cultivars of the subgroup Cavendish, which may explain the susceptibility of these genotypes to black Sigatoka [85,86]. Until recently, there has been no standardized protocol for studies of gene expression in bananas during interactions with *P. fijiensis*, which may be a contributing factor to differences in the results obtained. The characterization of the functionality of genes and proteins can facilitate the understanding of the functions of these genes in the host and thus the development of new disease control strategies.

In summary, the results discussed here may contribute to banana and plantain breeding programs and the development of crossbreeding strategies in addition to identifying the genes (ACC oxidase, LOX, PR-4, GDSL, and JAR1) related to the plant defence response. These genes can be used both in the selection of molecular markers and to develop genetically modified plants with the use of cisgenics or gene editing techniques based on CRISPR to obtain cultivars tolerant or resistant to black Sigatoka.

## 4. Material and Methods

### 4.1. Plant Material

The experiment was conducted in the experimental area of Embrapa Mandioca e Fruticultura, located in Cruz das Almas, Bahia, Brazil. Plantlets of four contrasting genotypes were used to evaluate the interaction of plants with the fungus *P. fijiensis*. The genotypes used were Calcutta 4 (AA diploid, BSresistant), Krasan Saichon (AA diploid, BSresistant), Grand Naine (AAA triploid, BS-susceptible) and Akondro Mainty (AA diploid, BS-susceptible). These genotypes were chosen based on phenotyping previously performed in an experimental field by Nascimento et al. [9], in which resistance and susceptibility to the disease were verified.

The plantlets were grown in vitro, acclimated in a greenhouse in trays for 38 days and then transplanted to 30 × 40 cm polyethylene bags containing a mixture of commercial substrate with a basis of *Pinus* bark (Tecnomax, Coronel Freitas, Santa Catarina, Brazil) and coconut fiber (5:1; *v*:*v*), where they remained for 6 months. A total of 60 plants were used in this study. The experimental design was completely randomized with 15 plants for each genotype and two treatments (pathogen-inoculated cultivar and non-inoculated cultivar); 12 plants inoculated with the pathogen and three controls. 

### 4.2. Fungal Material

The monosporic isolate of black Sigatoka was originally collected from the cultivar “Grand Naine” at the Embrapa Cassava and Fruits experimental area. Banana leaves with typical symptoms of black Sigatoka were taken to the Phytopathology Laboratory and washed in running water to remove impurities and contaminants. After rinsing, the leaves were kept in acrylic boxes containing moistened cotton for 48 h with the temperature adjusted to 25 ± 2 °C and a photoperiod of 12 h. Afterwards, with the aid of a binocular stereomicroscope, the sporulation of the pathogen was verified, and the fungal structures were collected.

The pathogen spores were carefully collected from the lesions using a fine-tipped needle and transferred to Petri dishes containing potato, dextrose and agar (PDA) medium. The plates were sealed and incubated in BOD (bio-oxygen demand) incubators at 25 °C ± 2 °C for 15 days. After incubation, the colonies of *P. fijiensis* isolates were macerated by adding 4 mL of sterile distilled water, forming a fungal suspension. Then, the suspension was distributed over the entire surface of Petri dishes containing PDA medium, and the plates were sealed and incubated for 10 days in the BOD incubators at ± 25 °C. After this incubation period, molecular diagnosis was performed according to [87] using species-specific primers, which confirmed that the fungus was *P. fijiensis*.

### 4.3. Plant Inoculation and Sample Collection

Inoculation was performed following the protocol described by Cordeiro et al. [88] with adaptations. Briefly, a suspension of the monosporic isolate at a concentration of 1 × 10^4^ conidia/mL was prepared using a Neubauer chamber to adjust the suspension or inoculum. At 6 months, leaves 1 and 2, which represent the first and second youngest expanded leaves, respectively, were inoculated in the greenhouse, where the suspension was brushed on the adaxial part of the leaves, a method suggested by Leiva-Mora et al. [89]. After inoculation, the leaves were covered with transparent plastic bags to maintain the humidity of the environment, favoring the development of the disease.

The first collection was performed at time 0 for the control (plants not inoculated with the pathogen). Samples were collected 3, 9, 15 and 21 days after inoculation. These intervals were selected according to the same sampling intervals available in the literature for this pathosystem, *Musa* sp. × *P. fijiensis* [23,52,90]. After collection, the samples were immediately immersed in liquid nitrogen and then stored in an ultrafreezer at −80 °C for subsequent processing. Three biological replicates of each treatment were used for each analysis.

### 4.4. Molecular Analysis

#### 4.4.1. Total RNA Extraction and cDNA Synthesis

RNA extraction was performed at the Advanced Biology Center of Embrapa Mandioca e Fruticultura. The cetyltrimethylammonium bromide (CTAB) method proposed by Zhao et al. [91], was used. The evaluation of RNA quantity and quality was performed by comparative analysis of the samples in 1% agarose gel in 1X TAE buffer and by spectrophotometry using a Nanodrop ND-2000 device (Thermo Scientific, Waltham, MA, USA). Next, the samples were treated with DNase (RNase TURBOfree-Ambion) following the manufacturer’s recommendations and stored in an ultrafreezer (−80 °C). For cDNA synthesis, total RNA treated with DNase was used according to the manufacturer’s instructions for the high-capacity RNA-to-cDNA Kit (Applied Biosystems, Waltham, Massachusetts, EUA). To verify the viability of the cDNA, conventional PCR was performed using the 25S primer.

#### 4.4.2. Primers Used and RT–qPCR

For this study, seven primer pairs were initially selected for gene expression analysis (Table 1). The primer sequences used in this study were obtained from studies conducted by Portal et al. [23], Rodriguez et al. [24] and Podevin et al. [92] (Table 1).

#### 4.4.3. Quantitative PCR Analysis

The RT–qPCR analysis of the genes involved in the response to black Sigatoka in banana plants were performed using the ABI 7500 Fast Real-Time PCR System (Applied Biosystems, Foster City, CA, USA). Primer efficiency tests were performed with 5 serial dilutions and 3 replicates. The results were evaluated by assessing the slope of the standard curve and R^2^. The expression analyses of the studied genes were conducted in technical triplicates for each leaf sample collected. The reaction consisted of 1 μL of cDNA, 0.3 μL of each primer (RF), 5 μL of SYBR Green PCR mix (Ludwig Biotech, Alvorada-RS, Brazil) and 3.4 μL of nuclease-free water, for a total volume of 10 μL in each reaction. The thermocycling conditions of the reactions were as follows: 50 °C for 2 min and 95 °C for 10 min; 40 cycles of denaturation at 95 °C for 15 s and annealing and primer extension at 58 °C for 1 min. The melting curve was determined after the end of the reaction cycles of each product which was amplified between 72 and 95 °C.

### 4.5. Histological Analysis

#### 4.5.1. Clarification and Leaf Staining

To visualize the fungal structures in the studied genotypes, leaf fragments were clarified and stained following the methodology proposed by Phillips and Haymann [93] with adaptations. Clarification was performed by immersing the leaf sections (1 cm^2^) in 10% KOH (potassium hydroxide) solution at room temperature for 48 h. Then, the KOH was discarded, the samples were washed with water to remove any remaining residues, and the samples were transferred to 1% HCl solution for 30 min. Afterwards, the samples were stained with 0.05% Trypan Blue in solution (2:1:1, lactic acid:glycerol:water) for 1 h. Immediately after the dye was discarded, the leaf fragments were immersed in lactoglycerol solution (2:1:1, lactic acid:glycerol:water) and microphotographed under a light microscope.

#### 4.5.2. Stomatal Density

Stomatal density was determined from the count of stomatal cells on the abaxial part of the leaf. Three fragments, each with an area of 1 cm^2^, were removed from the leaves of non-inoculated plants of each genotype. Soon after, the leaves underwent a clarification process and were observed under an optical microscope (20×). The number of stomata was counted in an area of 34,394 μm. The results were expressed as stomatal density (number of stomata/mm^2^). In addition, the distance between stomata was measured for each genotype using ImageJ software (Wayne Rasband da Research Services Branch (RSB), EUA) [94].

#### 4.5.3. Histochemistry

For histochemical analysis, leaf sections were fixed in Karnovsky solution (Karnovsky, 1965) for 48 h, dehydrated in an increasing ethanol series (30–100%) for 3 h each, infiltrated and embedded using the Historesin Kit (hydroxyethylmethacrylate; Leica Heldelberg, Germany). Polymerization of the resin was performed at room temperature for 48 h. Serial histological sections (8 μm) were obtained in a Leitz rotary microtome (Model 1516), placed on histological slides and stained. Ferric chloride dye [95] was used to detect phenolic compounds, and aniline blue + Lugol dye was used to evaluate the presence of callose [96]. The histological sections were analyzed and photographed under a BxS1 fluorescence microscope (Olympus Latin America Inc., Tokyo, Japan).

#### 4.5.4. Scanning Electron Microscopy (SEM)

Samples from 0, 9 and 21 DAI were dehydrated in an ethanol series and dried in a critical point apparatus (Leica EM CPD 030) using liquid CO_2_. Next, the samples were fixed on a metallic support (stubs) with double-sided carbon adhesive tape and metallized with gold in a JEOL Smart Coater DII-29010SCTR device. The observations and electron micrographs were performed with a JEOL JSM-6390LV scanning electron microscope in the electron microscopy laboratory of the Gonçalo Moniz Institute, FIOCRUZ, Salvador-BA.

#### 4.5.5. Statistical Analysis

To perform the relative quantification of gene expression, comparative quantification was performed using the pair-wise fixed reallocation randomization test [97]. Statistical analysis was carried out using the R V.2.4 (R Core Development Team, http://cran.r-project.org/, 12 February 2022) software [98]. Data of relative quantification obtained was tested as to distribution by the normality test of Shapiro–Wilks using the “ExpDes.pt” package. Since the data did not meet the analysis of variance assumptions, the significant differences were tested using the Kruskal–Wallis test combined with the Dunn multiple comparison test (*p* > 0.05) using the “dplyr”, “rstatix”, “ggplot2”, “ggpubr” and “ggthemes packages. In this case, only the statistical differences between the cultivars in each time period were evaluated. Hierarchical group analysis was also performed for each gene based on the heat map where the expression values were converted into colors by the “gplots” package. Analysis of variance was used to evaluate the measures of stomatal density and distance between stomata and the Tukey test at 5% probability used to compare averages using the “ExpDes.pt” e “ggplot2” packages. 

## Figures and Tables

**Figure 1 plants-11-01953-f001:**
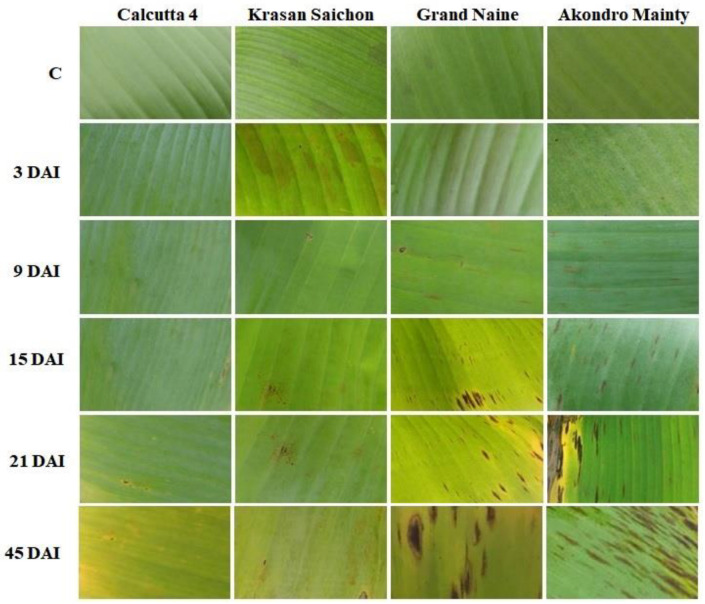
Progression of black Sigatoka symptoms in different banana genotypes inoculated in a greenhouse: Calcutta 4, Krasan Saichon, Grand Naine and Akondro Mainty. C: control; DAI: days after inoculation (3–45 DAI).

**Figure 2 plants-11-01953-f002:**
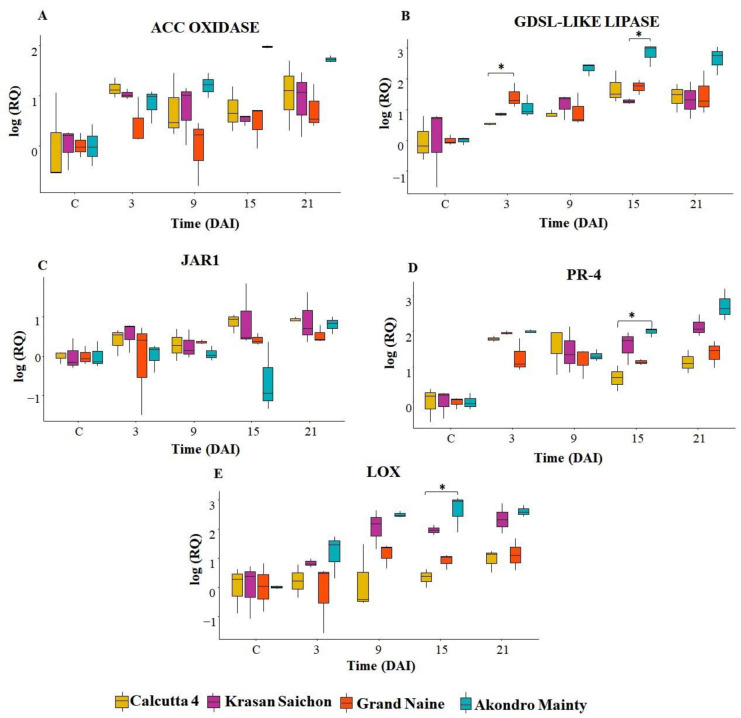
Box plots of the quantitative analysis of gene expression during the interaction of *P. fijiensis* and the banana cultivars Calcutta 4, Krasan Saichon, Grand Naine and Akondro Mainty. Relative expression levels (RT–qPCR) of the Acc oxidase (**A**), GDSL-like lipase (**B**), JAR1 (jasmonate resistant 1) (**C**), Pr-4 (**D**) and Lox lipoxygenase (**E**) genes. Quantifications were normalized using the reference gene 25S and tubulin. DAI: days after inoculation. Kruskal–Wallis with post hoc Dunn’s test was carried out to identify significant differences between the four cultivars within each time period * *p* < 0.05.

**Figure 3 plants-11-01953-f003:**
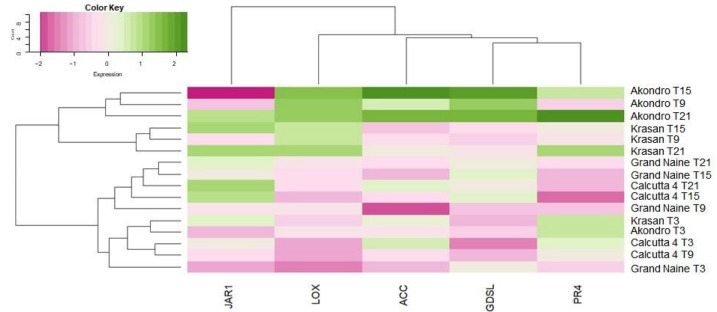
Hierarchical clustering of differentially expressed genes at four sampling times after inoculation with *P. fijiensis* in samples of Calcutta 4, Krasan Saichon, Grand Naine and Akondro Mainty. T3: 3 days after inoculation, T9: 9 days after inoculation, T15: 15 days after inoculation, T21: 21 days after inoculation. JAR1: jasmonate-amino acid conjugate enzyme; LOX: lipoxygenase; ACC: 1-amino-cyclopropane-1-carboxylate 1-carboxylic acid-1-aminocyclopropane; GDSL: lipase/hydrolase; PR4: pathogenesis-related protein. Quantifications were normalized using the 25S and tubulin *Musa* sp., reference genes.

**Figure 4 plants-11-01953-f004:**
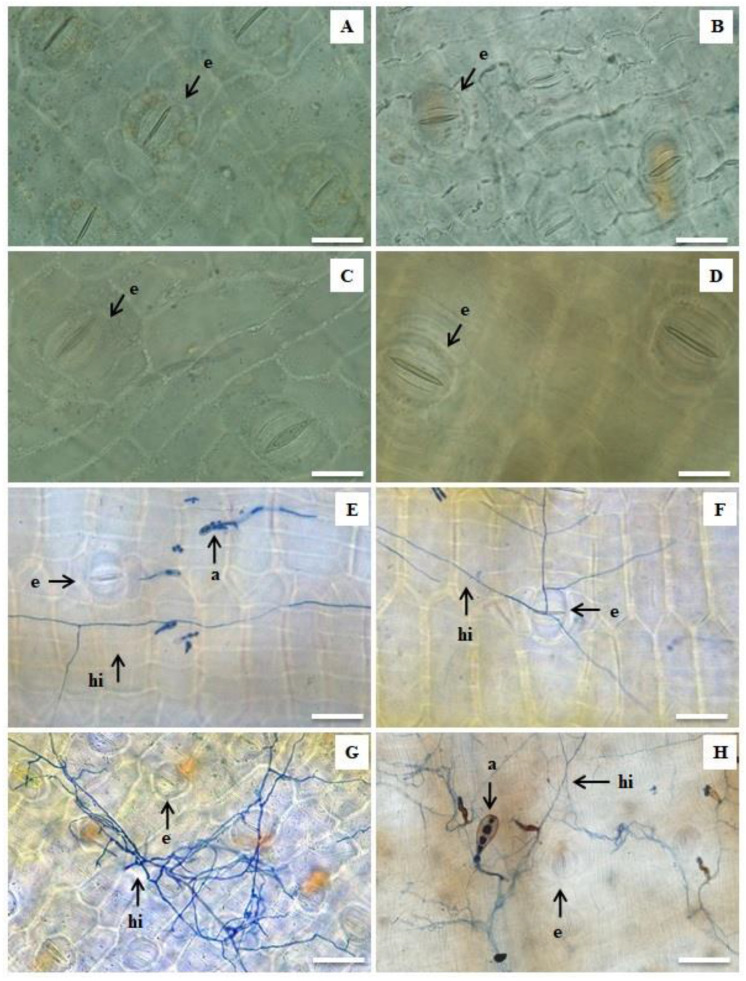
Clarification and staining of fungal structures in banana leaves. Non-inoculated treatments: Calcutta 4 (**A**), Krasan Saichon (**B**), Grand Naine (**C**) and Akondro Mainty (**D**). Treatments after inoculation (21 DAI) with *P. fijiensis* isolate: Calcutta 4 (**E**), Krasan Saichon (**F**), Grand Naine (**G**) and Akondro Mainty (**H**). a = ascospores; hi = hyphae; e = stomata. White bar (**A**–**D**): 50 μm; White bar (**E**–**H**): 200 μm.

**Figure 5 plants-11-01953-f005:**
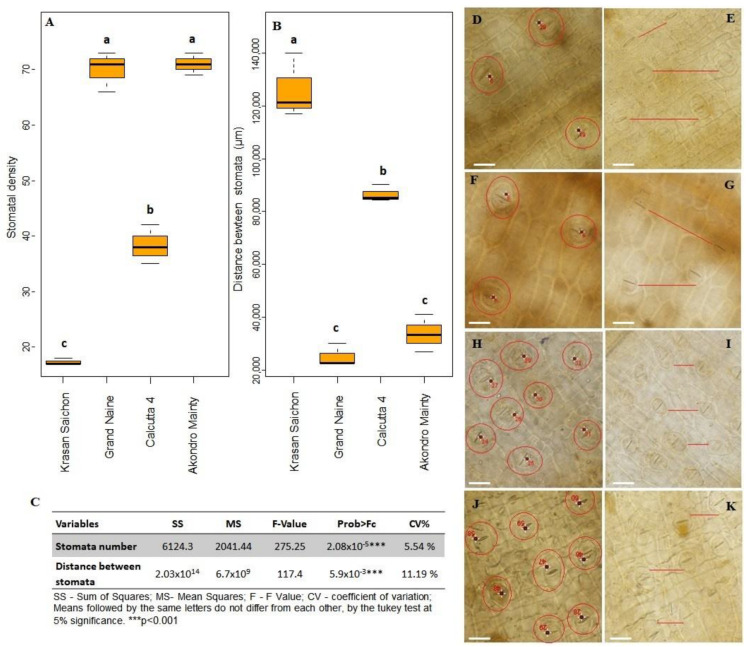
Stomatal analyses of banana genotypes evaluated for resistance to black Sigatoka: box plots of stomatal density (**A**) and distance between stomata (**B**); means followed by the same letter do not differ according to Tukey’s test at 1% significance. Counts of the number of stomata in measurements of distance between stomata in Calcutta 4 (**D**,**E**), Krasan Saichon (**F**,**G**), Grand Naine (**H**,**I**) and Akondro Mainty (**J**,**K**). Bar: 200 μm. Tukey’s test at 5% significance (**C**).

**Figure 6 plants-11-01953-f006:**
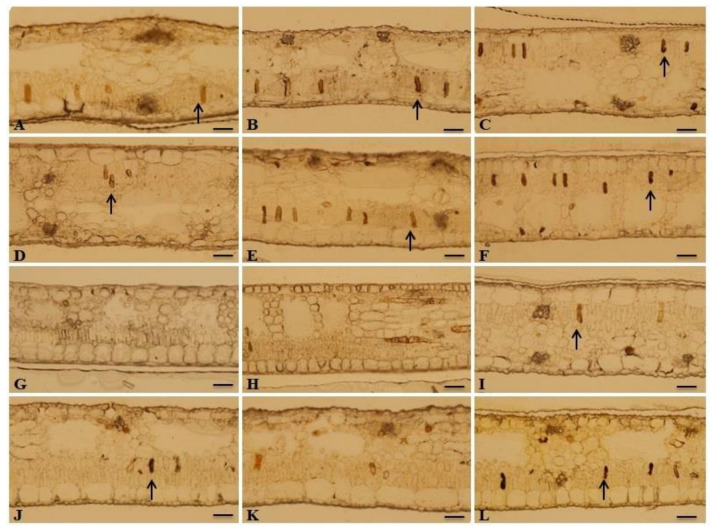
Cross sections of banana leaves of the genotypes Calcutta 4 (**A**–**C**), Krasan Saichon (**D**–**F**), Grand Naine (**G**–**I**), Akondro Mainty (**J**–**L**) inoculated with *P. fijiensis*. (**A**,**D**,**G**,**J**) = non-inoculated plants. The images are for 0, 9 and 21 DAI. Histochemical test with ferric chloride for the detection of phenolic compounds. Arrows indicate the production of phenolic compounds. Bar: 200 μm. Cruz das Almas-BA, 2022.

**Figure 7 plants-11-01953-f007:**
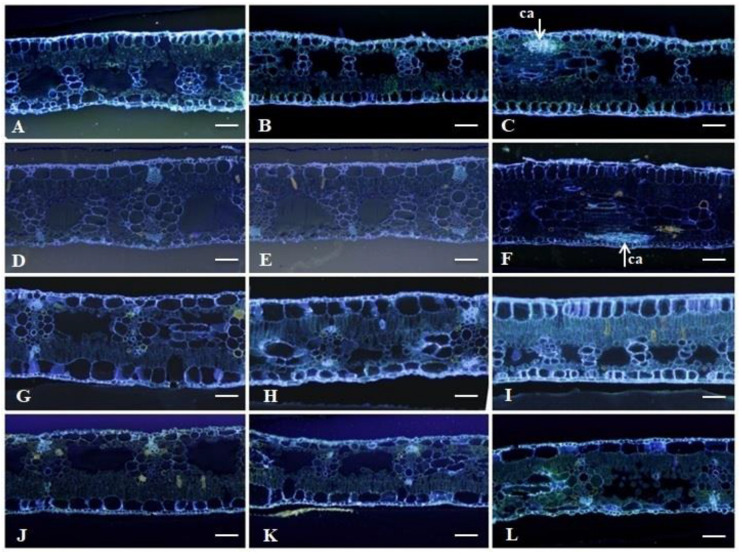
Cross sections of banana leaves of the genotypes Calcutta 4 (**A**–**C**), Krasan Saichon (**D**–**F**), Grand Naine (**G**–**I**), Akondro Mainty (**J**–**L**) inoculated with *P. fijiensis*. (**A**,**D**,**G**,**J**) = uninoculated plants. The images are for 0, 9 and 21 DAI, respectively. Histochemical test with aniline blue for callose detection. ca: callose deposition; bar: 200 μm. Cruz das Almas-BA, 2022.

**Figure 8 plants-11-01953-f008:**
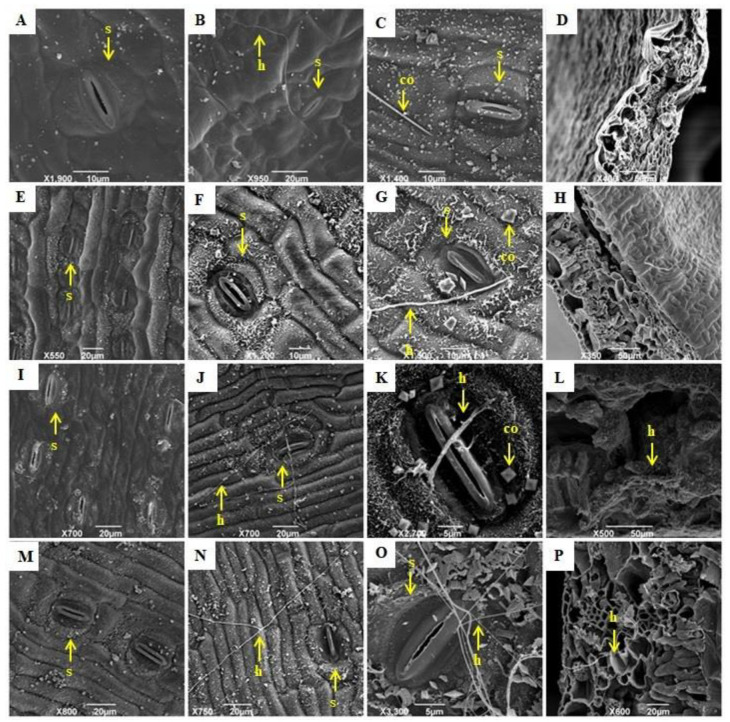
Scanning electron microscopy (SEM) was used to observe the interaction of *P. fijiensis* on the abaxial surface of leaves of the Calcutta 4 (**A**–**D**), Krasan Saichon (**E**–**H**), Grand Naine (**I**–**L**), Akondro Mainty (**M**–**P**), banana cultivars. The images were taken at 0 DAI (**A**,**E**,**I**,**M**), 9 DAI (**B**,**F**,**J**,**N**), and 21 DAI (**C**,**G**,**K**,**O**,**D**,**H**,**L**,**P**). h: Hypha; s: stoma; co: calcium oxalate crystals.

**Table 1 plants-11-01953-t001:** Primers used in this study to evaluate the gene expression of banana genotypes.

Gene	Sequence (5′-3′)	Amplicon (bp)	Reference
Acc oxidase	F: CATAAACACCGGAGACCAGA	114	[23]
	R: TTGTAGAAGGAAGCGATGGA		
Jar1	F: TCAACATCGACAAGAACACC	154	[23]
	R: CACAACTCCCAGAAGATCAC		
Pr-4	F: CAGGTACTTCTTCCTCCACT	100	[23]
	R: CTACTACCCGGCTCAGAATAA		
Gdsl-like lipase	F: GCGCATCACAAAAGAAGAAG	120	[23]
	R: CCCCGTCAACATTCATCAG		
LOX	F: ACTACTTGCTCTTCCCCAGCGG	120	[24]
	R: CGTCCCTCACACCACACGC		
25S	F: ACATTGTCAGGTGGGGAGTT	106	[92]
	R: CCTTTTGTTCCACACGAGATT		
TUB	F: TGTTGCATCCTGGTACTGCT	112	[92]
	R: GGCTTTCTTGCACTGGTACAC		

## Data Availability

All data were collected at Embrapa Mandioca e Fruticultura, Brazil.

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
