# Peer review of "Gene Expression, Histology and Histochemistry in the Interaction between *Musa* sp. and *Pseudocercospora fijiensis"

_plants, 2022, doi:10.3390/plants11151953_

Round 1
Reviewer 1 Report
The original article "Gene expression, histology and histochemistry in the interaction between Musa sp. and Pseudocercospora fijiensis" in my opinion addresses an interesting and current subject of relevant scientific interest. It is well in line with clear objectives, a plan and appropriate technical and scientific approaches. The introduction is subject-oriented and well supported by the bibliography. The discussion of the results is objective and the conclusions are strongly supported by the results. For all these reasons, my opinion is in favor of its publication in "Plants".
Author Response
We thank reviewer#1 for being favorable to the publication of our manuscript.

Reviewer 2 Report
This manuscript written Julianna et al. conducted an experiment to detect Musa sp. plant gene expression, histology and histochemistry changes in four different cultivars (including two resist ant and two susceptible genotypes) during the interaction with fungal pathogen Pseudocerospora fijiensis.
In general, this is an interesting study to give hints for future breeding programs to increase black Sigatoka resistance in banana plants.
The abstract can be more summarized to highlight the main findings, other than just listed the results from each experiment. The background part of this abstract is a bit long compare with the other parts (methods, results et al.) in the abstract.
It’s better to add statistics in Figure 2. It’s okay to have box plots to show the gene expression, but it’s important to show whether the effects of treatment (different cultivars) and/or time is significant.
Figure 3: it’s better to indicate what are “T3”, ”T9”, “T15” and “T21” in the legend.
It’s very effective to see the pictures in Figure 4. I was wondering how many leaves did you chose for the staining from each cultivar? And how did you select leaves from each cultivar?
The legends of Figure 6 and 7 are a bit confusing to me. Are panels A, B C for genotype Calcutta, or only A and C? Probably, it’s better to indicate banana genotypes in the panels. Which panels are the treatments with pathogen inoculation? Please clarify it better. Is the panel “K” missing?
For all nice pictures in Figures, are there any statistics to show the significancy? For instance, the proportion of certain traits were shown in plants, other than qualitative variables.
In Material and Methods 4.1: 15 plants for each genotype, how many plants were inoculated with pathogen? And how many plants were not inoculated with pathogen? Please clarify.
BOD incubator = Bio-Oxygen Demand incubator?
Statistical analysis: please clarify which functions were used in which R package for which analysis. Not just list the names of R packages.
Minor comments:
Did authors check the native phyllosphere microbiome of four plant genotypes, and their reactions/changes after pathogen inoculation? Probably, the native phyllosphere microbiome could also be a key factor for the resistance to pathogen invasion and black Sigatoka.
Author Response
Comments and Suggestions for Authors
- This manuscript written Julianna et al. conducted an experiment to detect Musa sp. plant gene expression, histology and histochemistry changes in four different cultivars (including two resist ant and two susceptible genotypes) during the interaction with fungal pathogen Pseudocerospora fijiensis. In general, this is an interesting study to give hints for future breeding programs to increase black Sigatoka resistance in banana plants.
Answer: We thank reviewer#2 for all the contributions to improve the manuscript.
- The abstract can be more summarized to highlight the main findings, other than just listed the results from each experiment. The background part of this abstract is a bit long compare with the other parts (methods, results et al.) in the abstract.
Answer: Modifications included in the text accordingly.
- It’s better to add statistics in Figure 2. It’s okay to have box plots to show the gene expression, but it’s important to show whether the effects of treatment (different cultivars) and/or time is significant.
Answer: Kruskal-Wallis test was carried out and included in the text along with Dunn´s post hoc text in order to identify the differences
- Figure 3: it’s better to indicate what are “T3”, ”T9”, “T15” and “T21” in the legend.
Answer: Adjustments made to Figure 3 accordingly.
- It’s very effective to see the pictures in Figure 4. I was wondering how many leaves did you chose for the staining from each cultivar? And how did you select leaves from each cultivar?
Answer: Three cuts from each genotype studyied was selected for the color analysis. This material was colected at the same time as the simple for RNA extraction.
- The legends of Figure 6 and 7 are a bit confusing to me. Are panels A, B C for genotype Calcutta, or only A and C? Probably, it’s better to indicate banana genotypes in the panels. Which panels are the treatments with pathogen inoculation? Please clarify it better. Is the panel “K” missing?
Answer: Typing errors, but the text was re-written for better understanding.
- For all nice pictures in Figures, are there any statistics to show the significancy? For instance, the proportion of certain traits were shown in plants, other than qualitative variables.
Answer: Figures were evaluated only at qualitative level, with no quantitative analysis since the initial objective was only to vizualize the presence or not of a certain compound or pathogen.
- In Material and Methods 4.1: 15 plants for each genotype, how many plants were inoculated with pathogen? And how many plants were not inoculated with pathogen? Please clarify.
Answer: The number of plants was clarified in the methodology in the text.
- BOD incubator = Bio-Oxygen Demand incubator?
Answer: Corrected in the text.
- Statistical analysis: please clarify which functions were used in which R package for which analysis. Not just list the names of R packages.
Answer: The functions of each R package were included in the text.
Minor comments:
Did authors check the native phyllosphere microbiome of four plant genotypes, and their reactions/changes after pathogen inoculation? Probably, the native phyllosphere microbiome could also be a key factor for the resistance to pathogen invasion and black Sigatoka.
Answer: The analysis in this manuscript were exclusively especific to the plant x pathogen interaction without taking into consideration the influcence of exteral factors such as microbiome of the phylosphere, that due to its complexity will be explored in future studies.

Round 2
Reviewer 2 Report
Thanks to authors for addressing all my comments accordingly. Most of my concerns were well addressed. For this version manuscript, I am in favor of publishing it in "Plants".